# Gains vs losses in pay-for-performance: Stated preference evidence from a U.S. survey

**Justin G. Trogdon**[1,2]*, **Aveena Khanderia**[1], **Kathryn Brignole**[2], **Jodi A. Lewis**[1], **Tara Licciardello Queen**[3]

1 Department of Health Policy and Management, Gillings School of Global Public Health, University of North Carolina at Chapel Hill, Chapel Hill, North Carolina, United States of America, 2 Lineberger Comprehensive Cancer Center, University of North Carolina at Chapel Hill, Chapel Hill, North Carolina, United States of America, 3 Department of Health Behavior, Gillings School of Public Health, University of North Carolina at Chapel Hill, Chapel Hill, North Carolina, United States of America

* justintrogdon@unc.edu

## Abstract

### Background

Pay-for-performance (P4P) incentives can be paid as a bonus (gain) or a penalty (loss). Diminishing marginal utility of wealth suggests that, starting from the same initial wealth, individuals dislike losses more than they like equivalent gains.

### Objective

This study reports the minimum financial gain or loss required to motivate primary care providers and clinical staff to try to increase their human papillomavirus (HPV) vaccination rates.

### Data

In 2022, we conducted a national U.S. survey through WebMD's Medscape Network of clinical staff working in primary care clinics that provided HPV vaccination to children ages 9 through 12 years (N = 2,527; response rate = 57%).

### Methods

We randomized respondents to one of two hypothetical HPV vaccine incentive designs: a bonus for reaching an unspecified target HPV vaccination rate and a penalty for failing to reach the unspecified target. The primary outcome is the self-reported smallest incentive amount (U.S. dollars) that would motivate participants to try and increase their HPV vaccination rates. We tested for differences across P4P designs using unadjusted responses and linear regressions adjusting for clinic and respondent characteristics. We also tested for heterogeneous responses by experience with incentizves, training, and rurality.

**Data availability statement:** Due to legal restrictions on grant-funded research, relevant data may be shared with a signed data use agreement through University of North Carolina's (Chapel Hill, North Carolina, USA) Office of Industry Contracting (http://research.unc.edu/oic/). Researchers wishing to access the data may contact OIC@unc.edu.

**Funding:** JT, KB, TQ are the authors who received the award. Research reported in this publication was supported by the National Cancer Institute of the National Institutes of Health under Award Number P01CA250989. The sponsors/funders did not play a role in the study design, data collection and analysis, decision to publish, or preparation of the manuscript. The content is solely the responsibility of the authors and does not necessarily represent the official views of the National Institutes of Health. https://www.nih.gov/about-nih/what-we-do/nih-almanac/national-cancer-institute-nci.

**Competing interests:** The authors have declared that no competing interests exist.

**Abbreviations:** HPV, human papillomavirus; P4P, pay for performance; CMS, Centers for Medicare and Medicaid Services; PA, physician assistant, APN, advanced practice nurse; RN, registered nurse, LPN/LVN, licensed practical/vocational nurse; MA, medical assistant; CNA, certified nursing assistant; CI, confidence interval.

## Results

The mean amount required to motivate effort was $2,155 in the gain P4P design and $1,185 in the loss P4P design (unadjusted difference = $970 [$p < 0.001$], adjusted difference = $967 [$p < 0.001$]). There were no heterogeneous effects by rurality or experience with incentives. Physicians reported the highest differences (in dollars) between gain and loss P4P designs.

## Conclusions

Stated preference data from primary care clinical staff suggests that effective P4P incentives could be half as large if designed as losses rather than gains.

## Introduction

Pay for performance (P4P), a common type of value-based payment, attaches financial incentives/disincentives to provider performance. It can steer primary care providers toward high value care by tying reimbursement to metric-driven outcomes, proven practices, and patient satisfaction [1]. Traditional economic utility theory and behavioral economics suggest that P4P incentive designs can further increase their effectiveness by using losses or penalties for their incentives [2]. Traditional utility theory assumes diminishing marginal utility of wealth in which each additional dollar yields slightly less additional utility than the previous dollar [3]. This implies that, starting at the same wealth, utility gains from an additional $X will be less than the utility loss from losing $X. Similarly, behavioral economics has demonstrated experimentally that incentives that yield the same wealth outcomes will be valued differently depending on whether the outcome came about via a gain or loss, with the pain of losing greater than the pleasure of gaining [4]. Therefore, the impact of a financial incentive delivered as a loss may be larger than if delivered as a gain [2,4]. The Centers for Medicare and Medicaid Services (CMS) has enacted several P4P models that use a loss design for incentives (e.g., Hospital Value-Based Purchasing Program, Hospital Readmissions Reduction Program, Hospital-Acquired Condition Reduction Program, Comprehensive Primary Care Plus) [1,5].

Although P4P, and the use of loss-based incentives, have been successful for other vaccines [6–14], their effect on provider and clinical staff motivation to increase human papillomavirus (HPV) vaccine uptake has not yet been established [14]. HPV vaccination is part of routine childhood immunization starting at age 9 and is given to prevent six types of HPV cancers [15]. However, in 2021, only 61.7% of adolescents aged 13–17 were up to date with their HPV vaccination, far below the Healthy People 2030 goal of 80% [15]. Parents' self-reported reasons for not getting HPV vaccine include concerns about safety, not getting a provider recommendation, thinking it is not needed, requiring more information, and believing their child is not yet due [16]. However, a strong provider recommendation can address most of these concerns. In recent survey data, children who received a provider recommendation were 28 percentage points (60%) more likely to get the HPV vaccine than children who did not receive a provider recommendation [17]. P4P schemes would incentivize strong provider recommendations, which is an important leverage point for increasing vaccination rates. Low vaccination rates, the ease of tracking vaccination, and clear definition of series completion make the HPV vaccine a vital and attainable quality improvement target for P4P programs [13,14,18–20].

Our team conducted a national survey of clinical staff involved in HPV vaccine delivery in the U.S. We randomized respondents to a hypothetical P4P financial incentive tied to an

unspecified HPV vaccine rate target achievement. Half of respondents' incentives were gains for reaching the target (i.e., financial bonus) and half of respondents' incentives were losses for failing to reach the target (i.e., financial penalty). This study reports the effect of gain vs loss incentive design on the self-reported minimum financial incentive amounts needed to motivate clinical staff behavior. The results will inform design of P4P models to improve HPV vaccine rates, and potentially other pediatric or immunization outcomes.

## Methods

### Survey participants

In May through July 2022, the Improving Provider Announcement Communication Training team conducted a national, online survey of clinical staff working in primary care clinics that provided HPV vaccine to children (N = 2,527; American Association for Public Opinion Research Response Rate #3 of 57%) [21]. WebMD Market Research recruited participants through their Medscape Network. Respondents were 1) certified to practice in the United States; 2) practiced as a physician, physician assistant (PA), advanced practice nurse (APN) including nurse practitioner, registered nurse (RN), licensed practical/vocational nurse (LPN/LVN), medical assistant (MA), or certified nursing assistant (CNA); 3) worked in pediatrics, family medicine, or general medicine specialties; and 4) had a role in HPV vaccination for children ages 9 through 12 years.

### Outcome variables

We randomized survey respondents to one of two hypothetical HPV vaccine incentive programs. Those randomized to a gain design saw: "Imagine you get an annual bonus for reaching a target HPV vaccination rate among your patients. What is the smallest bonus that would motivate you to try to increase your HPV vaccination rate?" Those randomized to a loss design saw: "Imagine your pay is lowered if you do not reach an annual target HPV vaccination rate among your patients. What is the smallest penalty that would motivate you to try to increase your HPV vaccination rate?" The primary outcome of interest was the responses to these questions, coded as a continuous variable in U.S. dollars. We winsorized the top one percent of responses by frame ($25,000 for the gain design and $10,000 for the loss design). The main explanatory variable was an indicator for which design the respondent received (loss [reference] vs gain).

### Covariates

Statistical models adjusted for characteristics of the clinic and clinical staff. Clinic characteristics included experience with any financial incentives in the past year (no [reference] or yes), specialty (pediatrics [reference], family medicine, or other), number of clinics in the healthcare system (not system [reference], 1–4 clinics, or 5 or more clinics), practice type (solo practice [reference], group practice, hospital or academic institution, federally-qualified health center or community health center, or other), number of providers (1 [reference], 2–5, 6–10, 11 or more physicians, PAs, or APNs), percent of children using Vaccine for Children (<25% [reference], 25–49%, 50–74%, 75–100%, or not sure), the number of patients ages 9–12 seen in a typical week (0–9 [reference], 10–24, 25–49, or 50+), and rurality (rural or non-rural [reference] determined using Rural-Urban Continuum Codes of the clinic's county). Respondent characteristics included training (physician [reference], PA, APN, nurse [RN, LPN, LVN], or assistant [MA, CNA]), gender (woman [reference], man, or nonbinary or other gender), race/ethnicity (White [reference]; Hispanic, Latino, or Spanish; Black or African American; Asian;

other or prefer not to say; or multiple), and years in practice (continuous). Race/ethnicity categories, representing race as a social construct, were self-reported and were included to explore potential differences in response to financial incentives.

## Statistical analysis

We first tested for covariate balance across the randomized designs using chi-square tests for categorical covariates and t-tests for continuous covariates. We also conducted a two-sample Kolmogorov-Smirnov test for equality of the distribution of the smallest self-reported incentives that would motivate behavior across designs. In the main analysis, we tested for differences in our outcome using linear regression with an indicator for design (loss [reference] vs gain) as the main explanatory variable. We hypothesized that clinical staff reported larger required incentives under a gain design than a loss design (i.e., the coefficient on the gain indicator will be positive).

To test whether the effect of gain vs loss designs differed by key characteristics, we estimated three separate linear regression models that included interactions between design and experience with incentives, training, and rurality, respectively. We hypothesized that clinical staff with prior P4P experience may give responses informed by past incentive amounts and that these actual incentives may have been smaller than desired, in retrospect, resulting in a smaller gap between gain and loss designs (i.e., the coefficient on the interaction would be negative). Regarding training, we hypothesized that clinical staff may conceptualize the required incentives relative to their salary levels. In that case, physicians would have the largest absolute difference between designs (i.e., the interactions between design and all other training categories would be negative). Finally, regarding rurality, we hypothesized that feelings of resource scarcity in rural clinics may increase risk aversion (i.e., the coefficient on the interaction between design and rurality would be positive).

We conducted two sensitivity analyses. First, we repeated the linear regressions above and included the other clinical and clinical staff characteristics described above as covariates. Second, 33 respondents (1.3%) reported seeing zero children aged 9–12 in a typical week. As these clinical staff may have few opportunities to vaccinate, we also repeated our main analyses dropping these 33 observations from the analysis sample.

We report robust standard errors. All statistical analyses were performed using Stata version 17.0 (College Station, TX). The study was approved by the University of North Carolina at Chapel Hill Institutional Review Board (IRB Study #21-2829). Participants provided written consent for the study.

## Results

Most of the clinical staff worked in non-rural clinics (90.8%) that were part of a system of 5 or more clinics (45.2%) and had 2–5 providers (39.0%) (Table 1). Respondents were majority women (71%), mostly physicians (48%), and had 15 years of experience on average. The distributions of all covariates except one were not significantly different across the randomized P4P design. The percentage of respondents who reported experience with incentives was slightly higher in the gain group (31.7% vs 28.1%, p = 0.049).

The mean amount required to motivate effort was \$2,155 under the gain design and \$1,185 under the loss design (difference = \$970, p < 0.001) (Table 2, Model 1). The median amount required to motivate effort was \$1,000 under the gain design and \$500 under the loss design (difference = \$500, p < 0.001). The distributions of incentives under the two designs were statistically significantly different (Kolmogorov-Smirnov p < 0.001).

When we stratified by rurality, the difference in means between gain and loss designs was \$946 (p < 0.001) among non-rural clinical staff and \$1,198 (p = 0.013) among rural clinical staff

**Table 1. Characteristics of the national survey participants.**

| | Gain (N = 1263) | Loss (N = 1264) | p-value |
|---|---|---|---|
| **Rural** | | | 0.675 |
| Non-rural | 1144 (90.6%) | 1151 (91.1%) | |
| Rural | 119 (9.4%) | 113 (8.9%) | |
| **Experience with incentives** | | | 0.049 |
| No experience with incentives | 863 (68.3%) | 909 (71.9%) | |
| Experience with incentives | 400 (31.7%) | 355 (28.1%) | |
| **Specialty** | | | 0.559 |
| Pediatrics | 552 (43.7%) | 560 (44.3%) | |
| Family medicine | 539 (42.7%) | 550 (43.5%) | |
| Other | 172 (13.6%) | 154 (12.2%) | |
| **Healthcare system** | | | 0.702 |
| Not system | 488 (38.6%) | 475 (37.6%) | |
| 1–4 clinics | 204 (16.2%) | 219 (17.3%) | |
| 5 or more clinics | 571 (45.2%) | 570 (45.1%) | |
| **Practice type** | | | 0.827 |
| Solo practice | 140 (11.1%) | 135 (10.7%) | |
| Group practice | 639 (50.6%) | 620 (49.1%) | |
| Hospital or academic institution | 245 (19.4%) | 267 (21.1%) | |
| FQHC or community health center | 133 (10.5%) | 139 (11.0%) | |
| Other | 106 (8.4%) | 103 (8.1%) | |
| **Number of prescribers** | | | 0.062 |
| 1 provider | 84 (6.7%) | 85 (6.7%) | |
| 2–5 providers | 494 (39.1%) | 492 (38.9%) | |
| 6–10 providers | 305 (24.1%) | 356 (28.2%) | |
| 11 or more | 380 (30.1%) | 331 (26.2%) | |
| **% of children using Vaccines for Children** | | | 0.230 |
| <25% | 366 (29.0%) | 316 (25.0%) | |
| 25–49% | 211 (16.7%) | 232 (18.4%) | |
| 50–74% | 216 (17.1%) | 216 (17.1%) | |
| 75–100% | 153 (12.1%) | 165 (13.1%) | |
| Not sure | 317 (25.1%) | 335 (26.5%) | |
| **Number of children ages 9-12 seen in typical week** | | | 0.649 |
| 0–9 | 369 (29.2%) | 361 (28.6%) | |
| 10–24 | 494 (39.1%) | 506 (40.0%) | |
| 25–49 | 262 (20.7%) | 244 (19.3%) | |
| 50+ | 138 (10.9%) | 153 (12.1%) | |
| **Medical training** | | | 0.975 |
| Physician | 616 (48.8%) | 607 (48.0%) | |
| Physician assistant | 98 (7.8%) | 100 (7.9%) | |
| Advanced practice nurse | 205 (16.2%) | 200 (15.8%) | |
| Nurse | 296 (23.4%) | 310 (24.5%) | |
| Assistant | 48 (3.8%) | 47 (3.7%) | |
| **Gender** | | | 0.346 |
| Woman | 914 (72.4%) | 896 (70.9%) | |
| Man | 305 (24.1%) | 332 (26.3%) | |
| Other | 44 (3.5%) | 36 (2.8%) | |

*(Continued)*

**Table 1.** (Continued)

| | Gain | Loss | p-value |
|---|---|---|---|
| | (N = 1263) | (N = 1264) | |
| **Race/ethnicity** | | | 0.688 |
| White | 836 (66.2%) | 828 (65.5%) | |
| Hispanic, Latino, or Spanish | 51 (4.0%) | 49 (3.9%) | |
| Black or African American | 60 (4.8%) | 63 (5.0%) | |
| Asian | 176 (13.9%) | 180 (14.2%) | |
| Other or prefer not to say | 87 (6.9%) | 103 (8.2%) | |
| Multiple | 53 (4.2%) | 41 (3.2%) | |
| **Years in practice** | | | 0.624 |
| Mean (SD) | 15.13 (10.86) | 15.35 (11.14) | |

(Table 2, Model 2). The difference between gain and loss designs was not significantly different from those working in rural and non-rural clinics; the interaction effect coefficient = $252 (95% confidence interval [CI] −$725, $1228). When we stratified by experience with financial incentives, the difference in mean required incentives between gain and loss designs was $1,096 (p < 0.001) among clinical staff with experience and was $909 (p < 0.001) among clinical staff without experience (Table 2, Model 3). The difference between gain and loss designs was not significantly different for those with and without experience with incentives ($187, 95% CI: −$339, $714). When we stratified by training, the highest difference in means between gain and loss was for physicians ($1,239, p < 0.001) and the lowest difference was for medical and nursing assistants (−$99, p = 0.789) (Table 2, Model 4). However, the relative difference between gain and loss for physicians (=1.86), advanced practice nurses (=2.01), and nurses (=1.86) were generally similar. The difference between gain and loss designs was $817 less (95% CI: −$1,559, −$74) for physician assistants than for physicians. The difference between gain and loss designs was $1,338 less (−$2,155, −$522) for nurse and medical assistants than for physicians.

In the sensitivity analyses, adjusting for covariates did not significantly change results. For example, gains were associated with $981 (95% CI: $736, $1226) more in required incentives than losses (S1 Table). Similarly, excluding 33 respondents who reported seeing zero patients aged 9-12 years in a typical week did not affect the results of the analysis (S2 Table).

## Discussion

In our survey, we quantified the amount of money required to motivate additional effort to increase HPV vaccination rates. Clinical staff randomized to a hypothetical P4P program tied to financial bonuses (gains) self-reported an amount about twice as high as the amount reported among staff randomized to a P4P program tied to penalties (losses). These results are consistent with predictions from traditional utility theory with diminishing marginal utility of wealth. Although our randomization design was not a direct test of loss aversion in behavioral economics, our results resemble studies that show that the pain of losing is psychologically about twice as powerful as the pleasure of gaining [4].

Our research also contributes to the literature by including the full clinical staff team in our survey, reflecting the collaborative nature of primary care and childhood vaccinations [22]. Our results suggest that framing amounts for incentives may differ by clinical role. We documented variation in required incentives among clinical staff with different training backgrounds. Physicians in our study tended to have the largest differences in their required incentive sizes between the gain and loss designs relative to other clinical staff. We did not see

**Table 2. Incentive Needed to Change Behavior ($): Gain vs. Loss.**

|  | Gain | Loss | Difference: Gain–Loss | p-value |
|---|---|---|---|---|
| **Model 1: Main effect** |  |  |  |  |
| Full sample | 2155 | 1185 | 970 | <0.001 |
|  | N = 1263 | N = 1264 | (124) |  |
| **Model 2: Rural stratification** |  |  |  |  |
| Rural | 2328 | 1130 | 1198 | 0.013 |
|  | N = 119 | N = 113 | (481) |  |
| Non-rural | 2137 | 1191 | 946 | <0.001 |
|  | N = 1144 | N = 1151 | (128) |  |
| **Model 3: Experience stratification** |  |  |  |  |
| Experience with incentives | 2311 | 1215 | 1096 | <0.001 |
|  | N = 400 | N = 355 | (223) |  |
| No experience with incentives | 2083 | 1174 | 909 | <0.001 |
|  | N = 863 | N = 909 | (150) |  |
| **Model 4: Training stratification** |  |  |  |  |
| Physician | 2679 | 1439 | 1239 | <0.001 |
|  | N = 616 | N = 607 | (190) |  |
| Physician Assistant | 1586 | 1164 | 422 | 0.197 |
|  | N = 98 | N = 100 | (328) |  |
| Advanced Practice Nurse | 1897 | 945 | 952 | 0.001 |
|  | N = 205 | N = 200 | (291) |  |
| Nurse | 1645 | 886 | 760 | 0.003 |
|  | N = 296 | N = 310 | (255) |  |
| Assistant | 844 | 943 | −99 | 0.789 |
|  | N = 48 | N = 47 | (371) |  |

Mean U.S. dollars reported for gain and loss designs with sample size. Difference (gain–loss) is the incremental difference from linear regression with full set of interactions between randomized gain vs. loss designs and rurality (Model 2), experience with incentives (Model 3), and training (Model 4). Robust standard errors of the gain – loss difference reported. p-values for tests of the null hypothesis that means for gain and loss are equal.

significant interactions between gain vs loss design and experience with incentives or rurality. These results could be useful for expanding P4P programs to include non-physician clinical staff, provided any incentivized action and outcomes are attributable to these staff and perceived as under their control as per P4P design best practice [13,20]. For incentive programs instituted and paid out at the organizational level, this data may be used by practice leaders to consider the full clinical care team in P4P implementation [23,24].

While P4P designed as a loss might be more effective than as a gain, it may be less feasible for at least three reasons. First, for loss-based incentives to work, participants must feel like the incentives are truly theirs to lose [25]. To this end, a payment system with the ability to charge money back from clinics would need to be in place. For example, CMS, one of the largest insurance payers with whom most clinics and hospitals contract, has alternative payment models with double-sided risk (i.e., participants can gain or lose money). Second, loss-based incentives are less popular with medical providers, which may limit participation in such P4P programs. For example, 67% of Accountable Care Organizations are enrolled in a Medicare Shared Savings Program with two-sided risk [26]. Additionally, penalties may discourage participation from providers who care for a large proportion of low-income patients, as these providers tend to perform worse in P4P programs [1]. Finally, loss-based incentives require

reliable and trusted measurement of outcomes. Such measurement can be difficult in cases with small sample sizes, as they are more prone to low precision of estimates and high risk of false positives or negatives [27].

An alternative hypothesis that could explain our findings hinges on the fact that our P4P program was hypothetical. It is possible that respondents were strategic by overstating the size of the bonus or understating the size of the penalty to achieve the best possible financial outcomes. Under this hypothesis, the gap between gain and loss designs required in an actual P4P would be smaller than we observed. While we cannot rule this out completely, we do not believe it is the sole explanation. In the extreme, the most "strategic" behavior would be to maximize the reported bonus ($100,000 was the limit in the survey) and minimize the reported penalty ($0). However, this is not what we observed. Only two respondents in the bonus design maxed out the response and over 10% reported $0. In the loss design, while over 10% reported the minimum of $0, over 10% also reported values at or above $5,000. Thus, we believe the stated preferences provide meaningful information.

This study had other limitations. The description of the P4P program in the scenario did not specify all required details (e.g., target HPV vaccination rate, patient inclusion/exclusion criteria to define the HPV vaccination rate). Therefore, respondents' answers may have been affected by their imagined details and answers may have been different under specific P4P details. Similarly, we did not observe respondents' income or risk aversion, which would be relevant to the extent of bonus or penalty required to motivate action. In our design, we rely on the randomization of the gain vs loss design to mitigate any bias in the analysis of the difference in responses. In other words, we expect that, like our measured covariates (Table 1), imagined P4P details, income and risk aversion will be balanced across the gain and loss design samples and, therefore, not correlated with the treatment of interest. Future research would benefit from direct measurement of these factors. Although our research encompassed a substantial and diverse group of clinical personnel on a nationwide scale, including a high response rate, it may not provide a complete reflection of the intended workforce in the United States. We compared the demographics of our participant pool sourced from the Medscape panel and the 2022 Annual Social and Economic Supplement of the Current Population Survey, categorized by occupation. While attributes like age, gender, and ethnicity were mostly similar, discrepancies arose in certain areas. Specifically, our sample tended to be slightly younger, with a higher proportion of female physicians, and a greater number of male advanced practitioners and nurses. Moreover, our sample comprised a lesser proportion of White physicians and advanced practitioners, while a higher proportion of White individuals was observed among nurses and nursing staff (available upon request).

This study's results provide new evidence about the importance of incentive design in P4P programs aimed at HPV vaccine. Payers and healthcare systems may find this information useful as they consider expanding P4P programs to target HPV vaccine rates and/or incorporate non-physician clinical staff. Stated preference data from primary care clinical staff suggests that effective incentives could be half as large if delivered as losses rather than gains.

## Supporting information

**S1 Table. Incentive Needed to Change Behavior ($): Gain vs. Loss Adjusted for Covariates.**
(DOCX)

**S2 Table. Incentive Needed to Change Behavior ($): Gain vs. Loss Dropping Respondents with Zero Children Seen.**
(DOCX)

**S3 File. Survey codebook.**
(XLSX)

## Author contributions

**Conceptualization:** Justin G. Trogdon.

**Data curation:** Kathryn Brignole, Tara Licciardello Queen.

**Formal analysis:** Justin G. Trogdon, Aveena Khanderia, Jodi A. Lewis.

**Funding acquisition:** Justin G. Trogdon, Tara Licciardello Queen.

**Investigation:** Justin G. Trogdon, Aveena Khanderia.

**Methodology:** Justin G. Trogdon, Tara Licciardello Queen.

**Project administration:** Kathryn Brignole.

**Supervision:** Justin G. Trogdon.

**Validation:** Jodi A. Lewis.

**Visualization:** Aveena Khanderia.

**Writing – original draft:** Justin G. Trogdon, Aveena Khanderia.

**Writing – review & editing:** Justin G. Trogdon, Aveena Khanderia, Kathryn Brignole, Jodi A. Lewis, Tara Licciardello Queen.

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
