## [Decision Letter · Decision Letter 0]

23 Oct 2024

PONE-D-24-33322Prospect Theory in Pay-for-Performance: Stated Preference Evidence from a U.S. SurveyPLOS ONE

Dear Dr. Khanderia,

Thank you for submitting your manuscript to PLOS ONE. After careful consideration, we feel that it has merit but does not fully meet PLOS ONE’s publication criteria as it currently stands. Therefore, we invite you to submit a revised version of the manuscript that addresses the points raised during the review process.

**ACADEMIC EDITOR:** Please refer to the work of the relevant authors instead of the website as a reference. Although the manuscript's title is Prospect Theory, the content is only about the losses and gains of the value function. Therefore, the title of the article does not fully reflect the content. Although one reviewer has rejected this manuscript and the other reviewer has made comments that would require rejecting the article, I would like to give you a chance to revise it, because the manuscript subject is important. 

We look forward to receiving your revised manuscript.

Kind regards,

Hatime Kamilcelebi

Academic Editor

PLOS ONE

Reviewers' comments:

Reviewer's Responses to Questions

**Comments to the Author**

1. Is the manuscript technically sound, and do the data support the conclusions?

Reviewer #1: Partly

Reviewer #2: Yes

Reviewer #3: Partly

2. Has the statistical analysis been performed appropriately and rigorously?

Reviewer #1: Yes

Reviewer #2: Yes

Reviewer #3: I Don't Know

3. Have the authors made all data underlying the findings in their manuscript fully available?

Reviewer #1: Yes

Reviewer #2: Yes

Reviewer #3: No

4. Is the manuscript presented in an intelligible fashion and written in standard English?

Reviewer #1: Yes

Reviewer #2: Yes

Reviewer #3: Yes

5. Review Comments to the Author

Reviewer #1: Comments to authors:

This is a well done survey study providing estimates of how much money, in reward or penalty, clinicians think it will take to motivate them to increase vaccination rates. I have a few major concerns however, that I explain below.

1. Not really testing prospect theory. Many people say that behavioral economics shows that losses loom larger than gains. But that phenomenon is consistent with traditional economics, too, because of the downsloping of utility curves. I’ll explain with an extreme example.

a. Give me $500K and it will feel great

b. Take away $500k and I’ll be devastated. The loss looms much larger than the gain because it occurs on the steeper part of my utility/well-being curve.

c. Behavioral econ added a twist on this idea. The SAME outcomes, when occurring by gain v loss, feel different. Give people $10 dollars and then take it away and they will feel differently than if you take away $10 and give it back. They all end up in the same place, but that final place feels different depending on whether you got there through a loss or a gain. I’m probably not explaining this well, but let me illustrate by discussing your study more.

2. Your main finding shows that a carrot has to be larger then a stick to equally motivate people, at least hypothetically. A true test of prospect theory would look like this:

a. Group 1 receives salary $X this year (whatever that is) but can receive up to $Y more if they achieve high vaccination rates. Now we see how motivating $Y is.

b. Group 2 gets a pay raise, to $X + Y. But they can get penalize up to $Y for low vaccination rates.

c. These are identical situations, financially speaking, but prospect theory would predict that Group 2 will be more motivated.

d. Yous study did not test this design. It compared

i. Group 1 gets $X + $Y

ii. Group 2 gets $X - $Y

3. I suggest you revise the paper to clarify this issue. Your study is important, if the finding holds up in practice. But it is more about bonus vs penalty than about prospect theory

4. In a revision, I’d also like to see discussion of feasibility. It is one thing to say a penalty is more effective than a bonus, dollar for dollar. But would that be politically/socially feasible? That is a whole other bit of behavioral economics.

5. Final small comment/question: you say physicians had a larger difference, bonus vs penalty, than others. But was that in absolute terms or relative? Was the percent difference larger, or did they just need bigger bonuses and bigger penalties, with the same ratio of bonus/penalty as other people?

Thanks for this interesting study

Reviewer #2: 

The study represent a large-scale survey of a diverse range of US medical providers involved in the administration of HPV vaccines to children ages 9 through 12. Providers reported on to their professional backgrounds (e.g., medical training, practice type, healthcare system, etc.) and indicated the extent of annual bonus (i.e., gain frame) or annual income penalty (i.e., loss frame) that would motivate them increase their HPV vaccination rate in accordance with an unspecified annual vaccination target. Experimental conditions were grounded in Prospect Theory and assumed that “losses loom larger than gains,” that is, that bonuses would have to be substantially larger than penalties to motivate an increase in HPV vaccine delivery.

The manuscript at hand bears several noteworthy strengths. First, the study draws on a large sample of US medical providers (N = 2527) from different areas and different professional backgrounds. This makes for a particularly diverse and representative sample. It is also noteworthy that the authors considered a range of relevant covariates, such as prior experience with incentive schemes. The manuscript is accessibly written and easy to read and screen.

Despite its many merits, below, I suggest changes that I believe would further improve the quality of the manuscript and its contents. I invite the authors to consider implementing these changes where feasible.

MINOR ISSUES

- Page 2: Within the first paragraph, prospect theory and the term “loss frame” (as week as its implied counterpart, “gain frame”) are introduced but barely elaborated on. I find the introduction much too superficial for a lay reader. I recommend that the authors briefly describe the core assumptions of the theory and give an example of what a loss versus gain frame may look like. This would support understanding of the “losses loom twice as large” effect observed later in the results.

- Page 2: The introduction fails to explain why HPV vaccination rates may be so much lower than desired and to which degree medical providers have the opportunity to bring up that percentage if desired by these providers. This must be known to understand how an incentive scheme may bring up vaccination numbers (i.e., if it is not within providers’ individual means, then the scheme cannot be successful). For example, is it a question of having to inform/convince parents? A question of having insufficient staff or time slots to provide the vaccine?

- Page 2: Specify that for the bonus and penalty, financial/income-based ones are meant.

- Page 3: The abstract describes the response rate as 57%, the methods section describes it as “3 of 57%.” It is not clear what these numbers, particularly the 3, correspond to – could this be clarified?

- Page 3: Specify US dollars are meant.

- Page 7: I find the following sentence hard to interpret, can this be reformulated? “First, P4P programs [should this be ‘program’?] participants may bend their attention closer to process adherence rather than care optimization”.

- The formatting of tables is very inconsistent.

MAJOR ISSUES

- Limitations: Based on the instructions on page 3, the scenario did not specify an absolute target rate (e.g., x children per week/year) or a relative one (e.g., +10% more children in a week/year than at the moment). Given that the number of vaccinations administered or possibilities to do so will differ considerably by participant, participants’ perception of the feasibility of an uncertain, potential target likely biased their responses. It should be noted as a limitation that it is not known what target participants may have thought of, or how a more certain target (e.g., + 10%, which would still allow for differences in current vaccination numbers between participants) would have influenced the results.

- Limitations: Table 1 indicates that around 30% of the sample sees as few as 0 children in the relevant age range per week. This suggests that a considerable percentage of the sample is rarely in a position to administer HPV vaccines and should be identified as a limitation. This variable also appears absent as a control for the regressions reported in Table 4, although many others are included as covariates. The authors should either justify the omission of this variable from the analysis or re-run the regression with this variable included as a covariate. Furthermore, the authors may want to add an appendix in which they re-run the core analyses without all of the participants who see 0-9 children of the right age per week, as these participants may simply not be a good fit for the sample population.

- Limitations: Although many demographic/training-related covariates are included, two core covariates – income and a measure of trait loss aversion or similar concept relevant to prospect theory – were not assessed. Both seem highly relevant to the extent of bonus/penalty tolerated. I would note that future studies would benefit from their inclusion.

- Analyses: I find the statistical approach taken – many stratified t-tests, followed by regressions – not ideal, as it unnecessarily inflates the numbers of tests run. It strikes me as a more resource-efficient approach to simply run progressively more comprehensive regressions (starting with just the condition as IV and adding additional predictors to avoid the need for individual t-tests). Alternatively, the number of some tests could be halved by making use of using ANOVAs (e.g., an ANOVA with condition as one factor and experience as a second factor, rather than running separate tests for people with and without experience). I would, however, agree that for the reader, the inclusion of USD sums and differences in the results reporting is helpful and should be maintained when comparing groups.

Reviewer #3: While I very much liked and am excited by the premise of this work, I paused my review quickly after noticing the extremely casual effort put into the writing, study development, and analyses. This is a major topic with real-world consequences, but it has been written with almost no depth in the review or uses of information. There are 15 references, which is not alone an indication of quality, but when you see the casual statements made throughout the article that lack evidentiary backing, it raises a massive amount of concern about the rigor of the work itself. I would only consider this article if it was reapproached as a serious literature contribution, with extensive but concise statements about there it fits, what work it builds from, and much stronger technical acumen in the description of concepts (the pop psychology descriptions of loss aversion, prospect theory, etc. are really concerning).

6. PLOS authors have the option to publish the peer review history of their article (what does this mean? ). If published, this will include your full peer review and any attached files.

**Do you want your identity to be public for this peer review?** For information about this choice, including consent withdrawal, please see our Privacy Policy .

Reviewer #1: No

Reviewer #2: No

Reviewer #3: No

---

## [Author Response · Author response to Decision Letter 1]

10 Dec 2024

We thank the reviewers for their thoughtful comments and suggestions. Below we address each comment (in italics) and note where in the manuscript changes were made. We believe the paper has improved as a result.

ACADEMIC EDITOR:

1. Please refer to the work of the relevant authors instead of the website as a reference.

We have updated citations to cite the underlying authors rather than websites.

2. Although the manuscript's title is Prospect Theory, the content is only about the losses and gains of the value function. Therefore, the title of the article does not fully reflect the content.

We have updated the title of the article to “Gain vs Loss Framing in Pay-for-Performance: Stated Preference Evidence from a U.S. Survey.”

Reviewer #1:

1. Not really testing prospect theory. Many people say that behavioral economics shows that losses loom larger than gains. But that phenomenon is consistent with traditional economics, too, because of the downsloping of utility curves. I’ll explain with an extreme example.

a. Give me $500K and it will feel great

b. Take away $500k and I’ll be devastated. The loss looms much larger than the gain because it occurs on the steeper part of my utility/well-being curve.

c. Behavioral econ added a twist on this idea. The SAME outcomes, when occurring by gain v loss, feel different. Give people $10 dollars and then take it away and they will feel differently than if you take away $10 and give it back. They all end up in the same place, but that final place feels different depending on whether you got there through a loss or a gain. I’m probably not explaining this well, but let me illustrate by discussing your study more.

Your main finding shows that a carrot has to be larger then a stick to equally motivate people, at least hypothetically. A true test of prospect theory would look like this:

a. Group 1 receives salary $X this year (whatever that is) but can receive up to $Y more if they achieve high vaccination rates. Now we see how motivating $Y is.

b. Group 2 gets a pay raise, to $X + Y. But they can get penalize up to $Y for low vaccination rates.

c. These are identical situations, financially speaking, but prospect theory would predict that Group 2 will be more motivated.

d. Your study did not test this design. It compared

i. Group 1 gets $X + $Y

ii. Group 2 gets $X - $Y

I suggest you revise the paper to clarify this issue. Your study is important, if the finding holds up in practice. But it is more about bonus vs penalty than about prospect theory

We thank the reviewer for their thoughtful comment and clear examples. We agree that our survey experiment focused on gain vs loss designs for P4P and was not a clean test of prospect theory. Therefore, we updated the title and revised the Introduction and Discussion to remove references to prospect theory as motivation for the paper. Instead, we now focus on using gain vs loss frames in P4P and cite traditional economic utility theory to motivate the hypothesis (pp. 3-4, 11).

2. In a revision, I’d also like to see discussion of feasibility. It is one thing to say a penalty is more effective than a bonus, dollar for dollar. But would that be politically/socially feasible? That is a whole other bit of behavioral economics.

We added a paragraph in the Discussion that focuses on feasibility of loss frames for P4P in practice.

New Text (p. 12): “While P4P framed as a loss might be more effective than a gain frame, it may be less feasible for at least three reasons. First, for loss-framed incentives to work, participants must feel like the incentives are truly theirs to lose.(25) To this end, a payment system with the ability to charge money back from clinics would need to be in place. For example, CMS, one of the largest insurance payers with whom most clinics and hospitals contract, has alternative payment models with double-sided risk (i.e., participants can gain or lose money). Second, loss framed incentives are less popular with medical providers, which may limit participation in such P4P programs. For example, 67% of Accountable Care Organizations are enrolled in a Medicare Shared Savings Program with two-sided risk.(26)”

3. Final small comment/question: you say physicians had a larger difference, bonus vs penalty, than others. But was that in absolute terms or relative? Was the percent difference larger, or did they just need bigger bonuses and bigger penalties, with the same ratio of bonus/penalty as other people?

When we reported a larger difference between bonus vs penalty for physicians, we were referring to the absolute difference in dollars. Based on the reviewer’s comment, we added a sentence noting that, for the training categories with significant differences between the gain (bonus) and loss (penalty) frame, the relative ratio of amounts were generally similar and between 1.86 and 2.

New text (p. 10): “However, the relative difference between gain and loss for physicians (=1.86), advanced practice nurses (=2.01), and nurses (=1.86) were generally similar.”

Reviewer #2:

We thank the reviewer for the accurate summary of the paper and noting its strengths.

MINOR ISSUES

1. Page 2: Within the first paragraph, prospect theory and the term “loss frame” (as week as its implied counterpart, “gain frame”) are introduced but barely elaborated on. I find the introduction much too superficial for a lay reader. I recommend that the authors briefly describe the core assumptions of the theory and give an example of what a loss versus gain frame may look like. This would support understanding of the “losses loom twice as large” effect observed later in the results.

This comment was in line with Reviewer 1’s first comment. Because our survey experiment focused on gain vs loss designs for P4P and was not a clean test of prospect theory, we updated the title and revised the Introduction and Discussion to remove references to prospect theory as motivation for the paper. Instead, we now focus on using gain vs loss frames in P4P and cite traditional economic utility theory to motivate the hypothesis (pp. 3-4, 11).

2. Page 2: The introduction fails to explain why HPV vaccination rates may be so much lower than desired and to which degree medical providers have the opportunity to bring up that percentage if desired by these providers. This must be known to understand how an incentive scheme may bring up vaccination numbers (i.e., if it is not within providers’ individual means, then the scheme cannot be successful). For example, is it a question of having to inform/convince parents? A question of having insufficient staff or time slots to provide the vaccine?

We appreciate this suggestion. We added a section describing current barriers to HPV vaccination and emphasizing the crucial role that provider recommendations play in HPV vaccination. We argue that P4P schemes can influence provider recommendations, which is one of the most important leverage points for increasing vaccination rates.

New text (pp. 3-4): “Parents’ self-reported reasons for not getting HPV vaccine include concerns about safety, not getting a provider recommendation, thinking it’s not needed, requiring more information, and believing their child is not yet due.(16) However, a strong provider recommendation can address most of these concerns. In recent survey data, children who received a provider recommendation were 28 percentage points (60%) more likely to get the HPV vaccine than children who did not receive a provider recommendation.(17) P4P schemes would incentivize strong provider recommendations, which is an important leverage point for increasing vaccination rates.”

3. Page 2: Specify that for the bonus and penalty, financial/income-based ones are meant.

We clarified at our first mention of bonuses and penalties that these refer to financial bonuses and penalties.

New text (p. 4): “We randomized respondents to a hypothetical P4P financial incentive tied to an unspecified HPV vaccine rate target achievement. Half of respondents received a gain frame (i.e., financial bonus) and half received a loss frame (i.e., financial penalty). This study reports the effect of framing on the self-reported minimum financial incentive amounts needed to motivate clinical staff behavior.”

4. Page 3: The abstract describes the response rate as 57%, the methods section describes it as “3 of 57%.” It is not clear what these numbers, particularly the 3, correspond to – could this be clarified?

We apologize for the confusion. The American Association for Public Opinion Research defines several different types of response rates. We are reporting their Response Rate #3. We capitalized the term to make it more obvious that it refers to a proper noun/definition.

New text (p. 4): “(N=2,527; American Association for Public Opinion Research Response Rate #3 of 57%)”

5. Page 3: Specify US dollars are meant.

We now define our currency as US dollars.

New text (p. 5): “The primary outcome of interest was the responses to these questions, coded as a continuous variable in U.S. dollars.”

6. Page 7: I find the following sentence hard to interpret, can this be reformulated? “First, P4P programs [should this be ‘program’?] participants may bend their attention closer to process adherence rather than care optimization”.

This sentence was intended to point out that P4P participants focus on exactly what is incentivized, which may be related to, but not identical with, optimal care. However, in the case of HPV vaccination, the care process that can be monitored, measured, and incentivized is identical to the desired care outcome, namely HPV vaccination. Thus, we removed the sentence from the paper.

7. The formatting of tables is very inconsistent.

We updated all tables and formatted consistently.

MAJOR ISSUES

1. Limitations: Based on the instructions on page 3, the scenario did not specify an absolute target rate (e.g., x children per week/year) or a relative one (e.g., +10% more children in a week/year than at the moment). Given that the number of vaccinations administered or possibilities to do so will differ considerably by participant, participants’ perception of the feasibility of an uncertain, potential target likely biased their responses. It should be noted as a limitation that it is not known what target participants may have thought of, or how a more certain target (e.g., + 10%, which would still allow for differences in current vaccination numbers between participants) would have influenced the results.

The reviewer is correct that the scenario did not define a target that had to be met to trigger the incentives. We agree that respondents may have imagined different targets, some more feasible than others, which would affect their answers. However, we do not expect any bias from the unspecified target to differ systematically between the randomized gain vs loss frames. Nevertheless, we followed the reviewer’s suggestion and added this to the limitations section.

New text (p. 13): “The description of the P4P program in the scenario did not specify all required details (e.g., target HPV vaccination rate, patient inclusion/exclusion criteria to define the HPV vaccination rate). Therefore, respondents’ answers may have been affected by their imagined details and answers may have been different under specific P4P details.”

2. Limitations: Table 1 indicates that around 30% of the sample sees as few as 0 children in the relevant age range per week. This suggests that a considerable percentage of the sample is rarely in a position to administer HPV vaccines and should be identified as a limitation. This variable also appears absent as a control for the regressions reported in Table 4, although many others are included as covariates. The authors should either justify the omission of this variable from the analysis or re-run the regression with this variable included as a covariate. Furthermore, the authors may want to add an appendix in which they re-run the core analyses without all of the participants who see 0-9 children of the right age per week, as these participants may simply not be a good fit for the sample population.

While roughly 30% of the sample saw 0-9 children aged 9-12 in a typical week, only 33 respondents (1.3%) reported seeing zero children in a typical week. We have addressed this in two sensitivity analyses. First, when we control for covariates listed in Table 1, we do include the “number of children ages 9-12 seen in typical week” variable (see Supplemental Table 1). Second, we estimated our main analyses removing the 33 respondents who reported seeing zero children aged 9-12 in a typical week (Supplemental Table 2). In both analyses, the results were not meaningfully different from our main analysis (p. 11).

3. Limitations: Although many demographic/training-related covariates are included, two core covariates – income and a measure of trait loss aversion or similar concept relevant to prospect theory – were not assessed. Both seem highly relevant to the extent of bonus/penalty tolerated. I would note that future studies would benefit from their inclusion.

We agree with the reviewer that income and risk aversion are two important omitted factors. We rely on the randomization of the gain vs loss frame to mitigate any bias in the analysis of the difference in responses. In other words, we expect that, like our measured covariates (Table 1), income and risk aversion will be balanced across the gain and loss frame samples. Nevertheless, we followed the reviewer’s suggestion and added this to the limitations section.

New text (p. 13): “Similarly, we did not observe respondents’ income or risk aversion, which would be relevant to the extent of bonus or penalty required to motivate action. In our design, we rely on the randomization of the gain vs loss frame to mitigate any bias in the analysis of the difference in responses. In other words, we expect that, like our measured covariates (Table 1), imagined P4P details, income and risk aversion will be balanced across the gain and loss frame samples and, therefore, not correlated with the treatment of interest. Future research would benefit from direct measurement of these factors.”

4. Analyses: I find the statistical approach taken – many stratified t-tests, followed by regressions – not ideal, as it unnecessarily inflates the numbers of tests run. It strikes me as a more resource-efficient approach to simply run progressively more comprehensive regressions (starting with just the condition as IV and adding additional predictors to avoid the need for individual t-tests). Alternatively, the number of some tests could be halved by making use of using ANOVAs (e.g., an ANOVA with condition as one factor and experience as a second factor, rather than running separate tests for people with and without experience). I would, however, agree that for the reader, the inclusion of USD sums and differences in the results reporting is helpful and should be maintained when comparing groups.

We have adopted the reviewer’s suggestion in the revised paper. We now use linear regression with robust standard errors for all hypothesis tests. Model 1 includes only the randomized frame (gain vs loss) as the explanatory variable. Model 2 adds rurality fully interacted with gain/loss frame. Model 3 includes the gain/loss frame fully interacted with an indicator for experience with incentives. Finally, Model 4 includes the gain/loss frame fully interacted with indicators for respondents’ training. We use Stata’s “margins” commands to recover predicted (expected) incentive outcomes and incremental differences between the gain and loss frames by levels of the stratifying variables.

Reviewer #3:

1. While I very much liked and am excited by the premise of this work, I paused my review quickly after noticing the extremely casual effort put into the writing, study development, and analyses. This is a major topic with real-world consequences, but it has been written with almost no depth in the review or uses of information. There are 15 references, which is not alone an indication of quality, but when you see the casual statements made throughout the article that lack evidentiary backing, it raise

---

## [Decision Letter · Decision Letter 1]

5 Jan 2025

PONE-D-24-33322R1Gain vs loss framing in pay-for-performance: Stated preference evidence from a U.S. surveyPLOS ONE

Dear Dr. Trogdon,

Thank you for submitting your manuscript to PLOS ONE. After careful consideration, we feel that it has merit but does not fully meet PLOS ONE’s publication criteria as it currently stands. Therefore, we invite you to submit a revised version of the manuscript that addresses the points raised during the review process.

**ACADEMIC EDITOR: ** **We can see that in the main text, some references changed numbers, but in the reference section, this change is not mentioned through tracking.** **We would be pleased if you revised the manuscript, taking into account the first referee's evaluation.**

We look forward to receiving your revised manuscript.

Kind regards,

Hatime Kamilcelebi

Academic Editor

PLOS ONE

**Journal Requirements:**

Reviewers' comments:

Reviewer's Responses to Questions

**Comments to the Author**

1. If the authors have adequately addressed your comments raised in a previous round of review and you feel that this manuscript is now acceptable for publication, you may indicate that here to bypass the “Comments to the Author” section, enter your conflict of interest statement in the “Confidential to Editor” section, and submit your "Accept" recommendation.

Reviewer #1: (No Response)

Reviewer #2: (No Response)

2. Is the manuscript technically sound, and do the data support the conclusions?

Reviewer #1: Partly

Reviewer #2: Yes

3. Has the statistical analysis been performed appropriately and rigorously?

Reviewer #1: I Don't Know

Reviewer #2: Yes

4. Have the authors made all data underlying the findings in their manuscript fully available?

Reviewer #1: Yes

Reviewer #2: Yes

5. Is the manuscript presented in an intelligible fashion and written in standard English?

Reviewer #1: Yes

Reviewer #2: Yes

6. Review Comments to the Author

**Reviewer #1:**  The revision was responsive and well done. With one exception, and that might be my fault. I explained on the first review that this study did not test gain v loss framing. The authors responded by dropping reference to prospect theory. But I think it is still a mistake to call this gain vs loss framing. That language is more appropriate when we are comparing to identical situations but merely framing them differently. You like a surgery with a 90% survival rate better than one with a 10% mortality rate, even those are two ways of framing the same thing.

In this study, the gain v loss conditions are not the same intervention with a different framing. They are different interventions. Better language would be something that emphasizes the relative impact of carrots vs sticks, of gains vs losses; but you should emphasize, so readers don’t make the same mistake, that you did not test gain v loss framing.

If that isn’t clear, I’m happy to elaborate.

**Reviewer #2:**  Dear Dr. Kamilcelebi,

Thank you for this renewed opportunity to review the article PONE-D-24-33322R1, now titled “Gain vs loss framing in pay-for-performance: Stated preference evidence from a U.S. survey”.

I appreciate the authors’ thorough responses to all editor and reviewer comments made and believe the manuscript has improved substantially from the revisions. Below, I offer my responses to these revisions made by the research team (omitting the original reviewer and editor comments). Beyond those, I have no additional comments beyond noting minor typos/inconsistencies in the newly-added sections (e.g., “loss framed” instead of “loss-framed”, p. 13).

1. We have updated citations to cite the underlying authors rather than websites.

Response to 1: I apologize but I cannot see where these changes have taken place because there are no tracked changes concerning the reference list. I can see that in the main text, some references changed numbers, but cannot verify whether citations now focus on authors rather than websites. I trust this is the case.

2. We have updated the title […].

Response to 2: I approve of this change.

3. […] we updated the title and revised the Introduction and Discussion to remove references to prospect theory as motivation for the paper. Instead, we now focus on using gain vs loss frames in P4P and cite traditional economic utility theory to motivate the hypothesis (pp. 3-4, 11).

Response to 3: I approve of this change.

4. We added a paragraph in the Discussion that focuses on feasibility of loss frames for P4P in practice. [..]

Response to 4: I approve of this change.

5. […] we added a sentence noting that, for the training categories with significant differences between the gain (bonus) and loss (penalty) frame, the relative ratio of amounts were generally similar and between 1.86 and 2.

Response to : I approve of this change.

6. […] Instead, we now focus on using gain vs loss frames in P4P and cite traditional economic utility theory to motivate the hypothesis (pp. 3-4, 11).

Response to 6: I approve of this change.

7. [..] We added a section describing current barriers to HPV vaccination and emphasizing the crucial role that provider recommendations play in HPV vaccination. […]

Response to 7: I approve of this change; this addition very nicely provided plausible rationale for the present experiment.

8. We clarified at our first mention of bonuses and penalties that these refer to financial

bonuses and penalties. […]

Response 8: This has been clarified.

9. […] We are reporting their Response Rate #3. We capitalized the term to make it more obvious that it refers to a proper noun/definition.

Response to 9: This has been clarified.

10. We now define our currency as US dollars.[…]

Response to 10: This has been clarified.

11. […] we removed the sentence from the paper.

Response to 11: I approve of this change.

12. We updated all tables and formatted consistently.

Response to 12: This has improved.

13. […] , we followed the reviewers suggestion and added this to the limitations section. […]

Response to 13: This has improved.

14. While roughly 30% of the sample saw 0-9 children aged 9-12 in a typical week, only 33

respondents (1.3%) reported seeing zero children in a typical week. We have

addressed this in two sensitivity analyses. First, when we control for covariates listed in Table 1, we do include the“number of children ages 9-12 seen in typical week”

variable (see Supplemental Table 1). Second, we estimated our main analyses

removing the 33 respondents who reported seeing zero children aged 9-12 in a typical

week (Supplemental Table 2). In both analyses, the results were not meaningfully

different from our main analysis (p. 11).

Response to 14: I appreciate this helpful clarification and the changes made.

15. […] we followed the reviewer’s suggestion and added this to the limitations section. New text (p. 13): “Similarly, we did not observe respondents’ income or risk aversion, which would be relevant to the extent of bonus or penalty required to motivate action. In our design, we rely on the randomization of the gain vs loss frame to mitigate any bias in the analysis of the difference in responses. In other words, we expect that, like our measured covariates (Table 1), imagined P4P details, income and risk aversion will be balanced across the gain and loss frame samples and, therefore, not correlated with the treatment of interest. Future research would benefit from direct measurement of these factors.”

Response to 15: I approve of this change.

16. […] We now use linear regression with robust standard errors for all hypothesis tests. Model 1 includes only the randomized frame (gain vs loss) as the explanatory variable. Model 2 adds rurality fully interacted with gain/loss frame. Model 3 includes the gain/loss frame fully interacted with an indicator for experience with incentives. Finally, Model 4 includes the gain/loss frame fully interacted with indicators for respondents’ training. We use Stata’s“margins” commands to recover predicted (expected) incentive outcomes and incremental differences between the gain and loss frames by levels of the stratifying variables.

Response to 16: I approve of this change.

17. Based on this Reviewer’s comments and those from the other Reviewers, we have removed the focus on prospect theory as a motivator and now focus on the practical issue of designing P4P’s incentives as gain or losses. We cite traditional economic utility theory to motivate the hypothesis that the level of incentives needed to motivate action could be smaller when delivered as penalties (pp. 3-4, 11). As for where this paper fits into the broader literature, we updated our Introduction to include additional citations and context, nearly doubling the total number of citations (pp. 3-4). In brief, there is very little peer-reviewed evidence regarding P4P incentives for HPV vaccination, which is where we see the main contribution of this paper. The existing evidence on P4P for other vaccinations has generally not compared bonus (gain) vs penalty (loss) schemes.

Response to 17: I approve of this change.

7. PLOS authors have the option to publish the peer review history of their article (what does this mean? ). If published, this will include your full peer review and any attached files.

**Do you want your identity to be public for this peer review?** For information about this choice, including consent withdrawal, please see our Privacy Policy .

Reviewer #1: **Yes: ** Peter Ubel

Reviewer #2: **Yes: ** Julia Nolte

---

## [Author Response · Author response to Decision Letter 2]

16 Jan 2025

We thank the reviewers again for their thoughtful comments and suggestions. Below we address each comment and note where in the manuscript changes were made.

ACADEMIC EDITOR:

1. We can see that in the main text, some references changed numbers, but in the reference section, this change is not mentioned through tracking.

We have tracked changes that were made to the References, in text and in the bibliography, between the original submission and the revised document.

2. We would be pleased if you revised the manuscript, taking into account the first referee's evaluation.

In the revised manuscript, we removed all references to “framing” and focused on the difference between designing P4P incentives as gains vs losses.

Reviewer #1: The revision was responsive and well done. With one exception, and that might be my fault. I explained on the first review that this study did not test gain v loss framing. The authors responded by dropping reference to prospect theory. But I think it is still a mistake to call this gain vs loss framing. That language is more appropriate when we are comparing to identical situations but merely framing them differently. You like a surgery with a 90% survival rate better than one with a 10% mortality rate, even those are two ways of framing the same thing.

In this study, the gain v loss conditions are not the same intervention with a different framing. They are different interventions. Better language would be something that emphasizes the relative impact of carrots vs sticks, of gains vs losses; but you should emphasize, so readers don’t make the same mistake, that you did not test gain v loss framing. If that isn’t clear, I’m happy to elaborate.

Thank you for your clarification. In the revised manuscript, we removed all references to “framing” and focused on the difference between designing P4P incentives as gains vs losses.

Reviewer #2:

1. We have updated citations to cite the underlying authors rather than websites.

Response to 1: I apologize but I cannot see where these changes have taken place because there are no tracked changes concerning the reference list. I can see that in the main text, some references changed numbers, but cannot verify whether citations now focus on authors rather than websites. I trust this is the case.

We have tracked changes that were made to the References, in text and in the bibliography, between the original submission and the revised document.

---

## [Editor Report · Decision Letter 2]

21 Jan 2025

Gains vs Losses in Pay-for-Performance: Stated Preference Evidence from a U.S. Survey

PONE-D-24-33322R2

Dear Dr. Trogdon,

We’re pleased to inform you that your manuscript has been judged scientifically suitable for publication and will be formally accepted for publication once it meets all outstanding technical requirements.

Kind regards,

Hatime Kamilcelebi

Academic Editor

PLOS ONE

---

## [Editor Report · Acceptance letter]

PONE-D-24-33322R2

PLOS ONE

Dear Dr. Trogdon,

I'm pleased to inform you that your manuscript has been deemed suitable for publication in PLOS ONE. Congratulations! Your manuscript is now being handed over to our production team.

Kind regards,

on behalf of

Assoc. Prof. Hatime Kamilcelebi

Academic Editor

PLOS ONE